# Hamstring Injury Prevention Program and Recommendation for Stride Frequency during Tow-Training Optimization

**Yusaku Sugiura [1],\*, Kazuhiko Sakuma [2], Shimpei Fujita [3] and Keishoku Sakuraba [4]**

1   Sport Science, Meikai University, Chiba 279-8550, Japan
2   National Track & Field Team, Sprint Head Coach, Korea Association of Athletic Federation, Seoul 05402, Korea; kzsakuma@juntendo.ac.jp
3   College of Health and Welfare, J.F. Oberlin University, Tokyo 194-294, Japan; fujita_s@obirin.ac.jp
4   Department of Sports Medicine, Juntendo University, Tokyo 113-8421, Japan; sakuraba@kf6.so-net.ne.jp
\*   Correspondence: yusaku@meikai.ac.jp

**Abstract:** (1) Background: Although innovations and improvements in towing systems have been available, tow-training method has not been considered favored in the training context. Tow-training may enable high stride frequency if hamstring injuries do not occur. The purpose of this study was to prevent hamstring injuries during supramaximal running and to optimize tow-training. (2) Methods: We investigated the relationship between the number of hamstring injuries that occurred during supramaximal running and the contents of the prevention programs that have been implemented, i.e., 4 years of the baseline programs and 12 years of the intervention. (3) Results: The incidence of hamstring injuries per 1000 sprinters was 57.5 for baseline and 6.7 for intervention. A significant difference was observed in the incidence of hamstring injury between the different combinations of prevention programs ($p < 0.01$). (4) Conclusions: Tow-training was optimized by (1) preventing hamstring injuries by combination of strength, agility, and flexibility training programs and (2) advising the sprinters to press the leg onto the ground as fast as possible to increase stride frequency and to prevent stride lengthening.

**Keywords:** tow-training; stride frequency; hamstring injury

## 1. Introduction

Supramaximal running velocity has been used in sprint training in order to increase stride frequency [1]. Tow-training has been employed for sprinters to assist supramaximal running velocity. Running velocity is determined by stride frequency and stride length. In maximal sprinting, stride frequency plays a more decisive role than stride length does [1]. In addition, the large negative power of the knee flexors (hamstrings) and the large positive power of the hip extensor contribute to high stride frequency, enabling the sprinter to run at higher speeds [2,3]. The mechanical power exerted by the hamstrings, which function in hip extension and knee flexion, becomes greater in the swing phase than it is when running at maximal speed [2,4]. To improve running performance, it is necessary to follow a particular program to acquire higher stride frequency by exerting great power in the bi-articular muscles of the hamstrings, which cross both the hip and the knee joints [5].

Repeated supramaximal running can result in high stride frequency [1]. In supramaximal running, sprinters can sprint at a supramaximal level [6]. Further, tow-training, a method of supramaximal running, can result in a high level of neuromuscular performance [1]. Therefore, supramaximal running is suggested for specific sprint training programs [5] for its possible effects in increasing stride frequency in sprinters.

Supramaximal running is used to improve sprinting performance through enhanced stride frequency [1,6]. In supramaximal running, runners sprint for a certain period of time while maintaining a supramaximal speed [6]. Supramaximal training often uses tow-training with an elastic cord or a rope and pulley [6,7]. Although innovations and

improvements in towing systems have been available, the tow-training method has not been considered favored in the training context [7].

The first reason is that, as Hicks [7] reports, in most studies stride length increases, and stride frequency decreases or remains steady in supramaximal running. Even in research using a towing device that may offer precise control of assistance compared with other modes of overspeed training, a significant increase was observed in stride length but not stride frequency [8]. This may suggest that supramaximal running may not provide a transfer any greater than that provided by standard maximal running [9]. The second reason is that hamstring injuries tend to occur during tow-training [7,10]. In an experimental study, Mero et al. [10] reported that one of nine male sprinters suffered severe injury to the hamstring during supramaximal running. Researchers and coaches have highlighted issues related to hamstring injuries [7,10–12].

As Devlin [13] suggested, there are risk factors leading to an injury, and some factors may be more predictive than others. Agre [14] reported that most hamstring injuries occur while running or sprinting. He also listed several possible etiological factors that relate to the occurrence of a hamstring injury, such as inadequate flexibility of the muscles, inadequate muscle strength, and dyssynergic muscle contraction. According to Sugiura et al. [15], for sprinters with high sprint performance, not only strength deficits but also the lack of neuromuscular control and flexibility contributes to the occurrence of hamstring injuries. How strength, agility, and flexibility training induces the occurrence of hamstring injuries in sprinters has not been studied.

If strength, agility, and flexibility do not function at a high-enough level to allow running velocities that well exceed 10 m/sec [1,5,6,10,16–18], the risk of hamstring injury in tow-training involving repeated episodes of supramaximal running increases. To develop a prevention program for hamstring injury during supramaximal running, we described a systematic preventive method involving multiple factors that have a causal relationship and in the light of which it is necessary to evaluate the effectiveness of the intervention program. We investigated the relationship between the number of hamstring injuries that occurred during supramaximal running and the content of the prevention programs that we implemented. Eliminating hamstring injuries that occur during supramaximal running may enable maximal effectiveness of tow-training to increase stride frequency.

Tow-training may enable high stride frequency if hamstring injuries do not occur. In this study, we sought to prevent hamstring injuries during supramaximal running and to safely and effectively optimize tow-training.

## 2. Materials and Methods

### 2.1. Subjects

The study subjects were 386 collegiate male sprinters (aged 18–24 years) over the course of 16 track and field seasons. Some of them were top-ranked sprinters at intercollegiate or Japanese national championships. Among them was a member of the Japanese 4 × 100 m relay team (fourth place overall) at the XXVIII Olympic Games (2004, Athens) and a member of the Japanese 4 × 100 m relay team (second place overall) at the 23rd Universiade Games (2005, Izmir). Those whose running skills were low and had recent medical history (less than 6 months) were excluded.

All subjects were voluntary participants from the same track and field sprint team. The sprint training program was supervised by the same coach for each of the 16 seasons. The coach, a co-author of this paper (Mr. Sakuma, formerly from Juntendo University) also had the final decision on sprinters' participation in training and at meets. Complete records were maintained on training and meet data for each sprinter. This research study was approved by the institution. Informed consent was obtained from the subjects before their participation.

### 2.2. Study Design

Figure 1 presents the study design. The research study took place over 16 track and field seasons, from 1996, when supramaximal running was well established as a specific program for sprinters on our team, to 2011. Supramaximal running was first employed on our sprint team in 1988. To date, we and our co-researchers have conducted a study on safety and effectiveness of tow-training [5,6,16,17] and have worked to spread the prescription of tow-training throughout Japan.

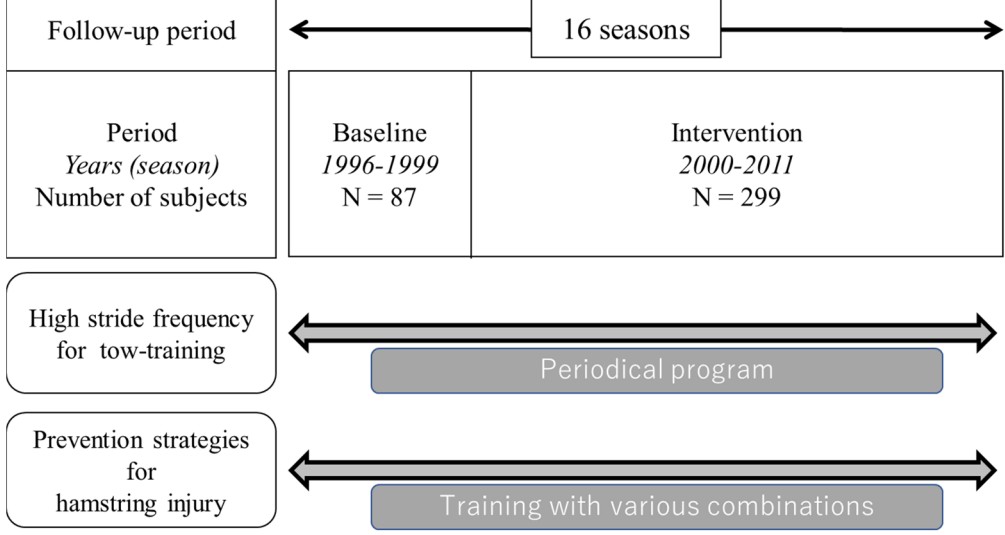

**Figure 1.** Study design.

The 16 seasons were divided into a baseline period of 4 years (1996–1999) and an intervention period of 12 years (2000–2011). Over this time, as is typical in athletics, new training methods and machines were developed and introduced, particularly in Olympic years [19].

### 2.3. Supramaximal Running

In tow-training, the velocity exceeds what can be achieved by the athlete in maximal sprinting. In long-term supramaximal running, it is possible to achieve changes in the nervous system that allow the neuromuscular performance to adapt to the higher speed level [1]. The purpose of this training is to improve sprinting ability by producing a higher stride frequency.

In our training, towing was performed using a towing machine [5,6,8,17] and a rubber tube [1,7] (Figure 2). The towing machine used allowed sprinters to be towed for ≥100 m. The rubber tube allowed many sprinters to be towed for up to 50 m.

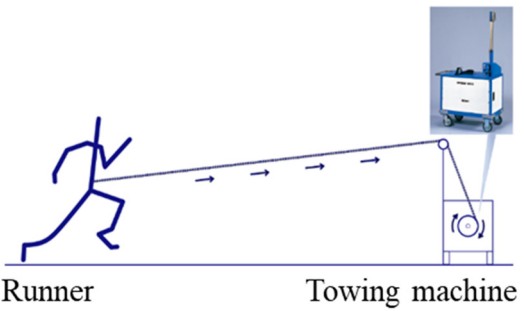

**Figure 2.** Supramaximal running in a towing system. The towing machine comprises an electric motor, a powder clutch, and a winch to wind rope. The powder clutch stabilizes the tension to pull the runner for safety.

Figure 3 presents the standard program prescribed for tow-training. Supramaximal running fell under the category of quality training in the sprinters' regimen; therefore, it was conducted at the point in the overall sprint training schedule when quality was being improved or when quality was high. During this period, the total intensity of training decreased. Tow-training was limited to the period from March to October, during which temperatures at the study site were ≥15 °C.

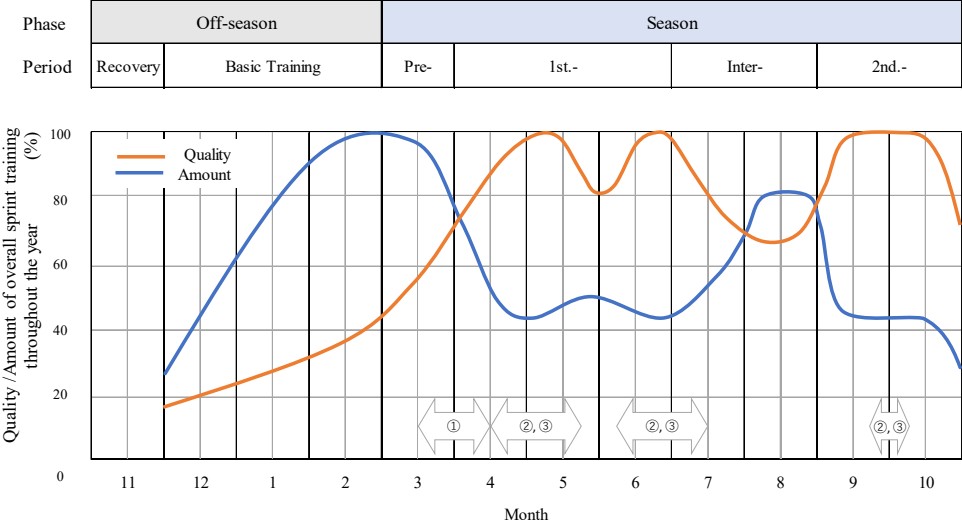

**Figure 3.** Prescription of standard program for tow-training. Feature: specific program (over-speed) for sprint training. Purpose: adapt the neuromuscular performance to high level/high stride frequency. Objectives: ① to increase maximal sprinting, ② to keep maximal sprinting, ③ to prepare for competition. Contents: ① ~105%; 50~100 m × 3~5; ② 105~110%, 50 m × 3; ③ 105~110%, 50 m × 1; represents the running velocity against maximal running. Distance and numbers are times of tow-training.

The higher levels of neuromuscular activity achieved by supramaximal running and the related higher stride frequency can improve the level of maximal sprinting, help maintain it, and even prepare for competition [1]. Supramaximal running to improve maximal sprinting took place at the beginning of each season, and those to maintain maximal sprinting and to prepare for competition took place during eachseason.

The percentual increase in the velocity of a sprinter during supramaximal running is up to 110% [1,5–7,9,10,16–18]. During our training, the rate was oftentimes set at 103–107%. Within this range, the sprinter could acquire a high stride frequency in a subjective sprinting motion [5,6,16]. Most importantly, the sprinters in this study were instructed to practice active landing, pressing the landing leg as quickly as possible onto the ground [1]. We had explained the purpose of tow-training to the sprinters to allow them to practice (implement) supramaximal running with a consciousness of higher stride frequency. To verify whether subjects follow this instruction, test runs for supramaximal running were executed.

Because muscle fatigue is a causative factor for hamstring injury [14], supramaximal running was performed on the day following a day off or on an individual practice day, when muscle fatigue was expected to be the lowest. Each sprinter engaged in 2–5 runs/day for 15–25 days/season. The number of runs per sprinter over a season was limited to a maximum of 50. In reality, none reached 50 runs; their number was from 20 to 30. The supramaximal running was adjusted appropriately so that it could be practiced on individualized principles.

### 2.4. Hamstring Injury Prevention Program

To provide an appropriate prevention program, the coach modified the program through repeated trial and error while investigating causative factors. As a result, the

purpose of the training was to improve neuromuscular function, muscle strength, and dynamic flexibility.

Table 1 presents the standard training program that the coach judged to be the most effective in preventing hamstring injuries. For the baseline phase, a traditional weight machine used for concentric knee flexor exercise (e.g., leg curl) and for concentric hip extensor exercise (e.g., hip extension) was recommended, and agility training, such as ladder and mini hurdle exercises, was incorporated into the program. In intervention, in addition to the programs that were used for baseline, eccentric strength body-weight exercise (e.g., lower Nordic hamstring exercise [20,21], glute-ham raise [22]), and dynamic stretching exercise) was employed.

**Table 1.** Description of the standard preventive program for hamstring injury.

| Objective and Method | Action and/or Motion (Load) | Period | |
|---|---|---|---|
| | | Baseline | Intervention |
| Strength | | | |
| Weight machine | · Knee flexors concentrically (leg curl) (3/5-4/5 of body weight by 10 repetitions × 3–5 sets) | • | • |
| | · Hip extensors concentrically (hip extension) (4/5-5/5 of body weight by 10 repetitions × 3–5 sets) | • | • |
| Body weight | · Knee flexors eccentrically (Nordic hamstring exercise) (lean forward slowly by 30–60 s × 5 sets) | | • |
| | · Knee flexors eccentrically and hip extensors/knee flexors concentrically (glute-ham raise) (lean forward, downward, and upward by 10–20 repetitions × 5 sets) | | • |
| Agility | | | |
| Ladder | · 5 types of fast stepping in all directions (10 m by 4 repetitions) | • | • |
| Mini-hurdle | · 4 types of one and/or both leg(s) with fast stepping (10 hurdles by 4 repetitions) | • | • |
| Flexibility | | | |
| Dynamic stretching | · 3 types of stretching for muscles around hip joint (20 m by 1 repetition) | | • |

The sprinters practiced the program in compliance with the load, action, and motion designated for each training method as mentioned in Table 1. The specific training used was adjusted in each case according to the judgment of the coach and with sufficient consideration for the condition of the given sprinter. Strength training was performed as a part of the weight training. Agility and flexibility training was performed during individual warm-ups.

In order to confirm whether hamstring injury prevention program was effective, the coach and the sprinters periodically investigated the effects of the prevention program. Muscle strengthening was objectively verified by increased ability (weight, repetitions, time, and the number of sets) in weight training. Neuromuscular function improvement was objectively verified by the decreased time required for each sprinter to clear 15 mini hurdles at the lowest height established at fixed intervals. Dynamic flexibility (measure of resistance to active motion around a joint or series of joints) [23] was determined by whether the sprinting motion was performed smoothly based on the subjective opinion of the coach and the sprinters themselves. Currently, no objective measurements have been established for dynamic flexibility as its relationship with goniometers, flexometers, and arthrometers is unproven [24].

*2.5. Recording Hamstring Injury*

Hamstring injuries were diagnosed by local tenderness, pain, and reduced range of motion on the straight leg raise test, as well as by evaluating pain and reduced strength during resisted knee flexion while prone [25]. We defined an incident of hamstring injury

as one due to which the sprinter had to withdraw for at least a week from training or competition [26].

### 2.6. Statistical Analysis

The number of stride frequency during maximal and supramaximal running was investigated in test runs. The difference in stride frequency between maximal and supramaximal running was compared using the Friedman's test.

The numbers of sprinters who had a hamstring injury and those who did not were aggregated for the baseline and intervention data. Next, the number of cases of hamstring injury was calculated in relation to different combinations of prevention programs. Injury incidence was calculated as the numbers of hamstring injuries per 1000 sprinters [27–29]. The difference between the varieties of prevention programs and hamstring injury incidence was investigated using a chi-squared test. The effect size (ES) and power in the post hoc tests were calculated using the GPower software (version 3.1). The ES between baseline and intervention was calculated using ES(w). The evaluations of ES strength were grouped into small (0.10), moderate (0.30), and large (0.50). The α error was set to $p < 0.05$, and the β error was set to $(1 - β) > 0.80$.

## 3. Results

### 3.1. The Number of Stride Frequency during Supramaximal Running

There were 10 subjects in the test runs for supramaximal running. The median of the stride frequency (strides/sec) in maximal running was 4.31 (range: 4.01–4.45), supramaximal running up to 105% was 4.40 (range: 4.09–4.76) and supramaximal running up to 110% was 4.55 (range: 4.26–4.76). The stride frequency during supramaximal running increased compared with maximal running ($p < 0.05$).

### 3.2. Incidence of Hamstring Injury

Figure 4 shows different incidences hamstring injury between baseline and intervention. The incidence of hamstring injuries per 1000 sprinters was 57.5 for baseline and 6.7 for intervention. A significant difference was observed in the incidence of hamstring injury according to the different combinations of prevention programs ($χ^2$ (2) = 7.11, $p < 0.01$, ES(w) = 0.21, $1 - β = 0.94$). The number of hamstring injuries was observed to decrease through a combination of prevention programs.

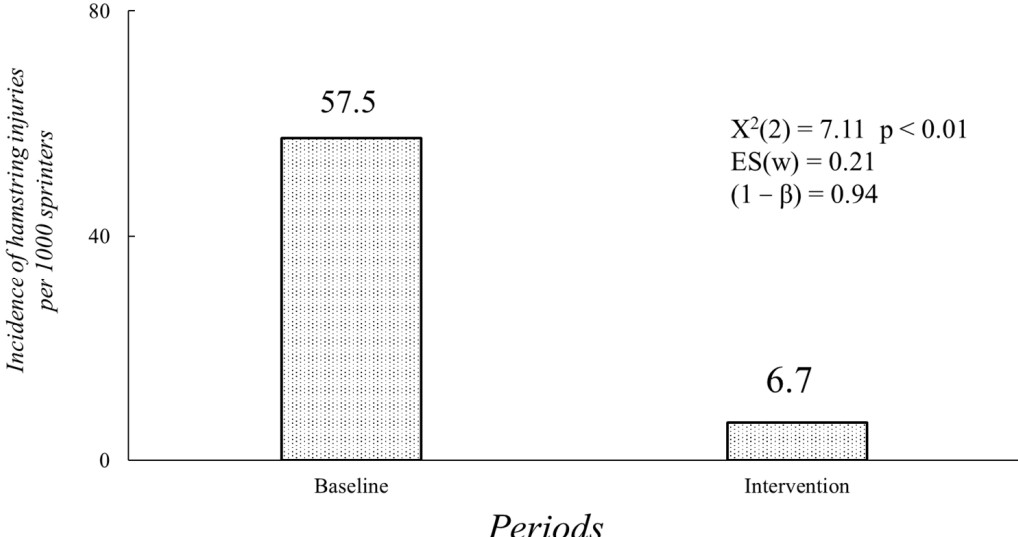

**Figure 4.** Effects of a combined prevention program on the incidence of hamstring injuries.

## 4. Discussion

To implement the safe and effective tow-training, this study observed a high-level collegiate sprint team for 16 years. The combination of prevention programs showed positive results. In terms of the effectiveness, the importance of active landing was confirmed.

### 4.1. Consideration of the Methodology

It was important for the quality of this study that one coach was leading the university track and field team (that is, the subjects of the study) for 16 years, during which the team remained competitive at the national level. Thanks to this fact, the 386 sprinters investigated can be treated as being of the same quality. In addition, the sprint training that employed supramaximal running was of excellent quality.

### 4.2. Recommendations for a Specific Program to Increase Stride Frequency

During the tow-training, Mero and Komi [1,18] emphasize moving the leg toward the ground as quickly as possible. This active landing is encouraged for two reasons. One is to prevent over-striding (an over-extended knee-joint position), a position that can lead to hamstring injury [30], and the other is to increase stride frequency [1,18].

According to this study, most sprinters were advised to press the leg as fast as possible onto the ground to increase stride frequency, and they obviously succeed well thanks to the advice. Nevertheless, in 3 out of 20 cases, the stride frequency decreased, and stride length was increased. This could have been affected by physical strength characteristics of the sprinter (the proportion of fast-twitch) [1] and running skills (running style) [6]. It is probably fair to say that these phenomena are applicable to all sprinters.

### 4.3. Effects of Agility Training on Hamstring Injury Incidence

From the late swing phase to the early contact phase in running at full speed, the hamstring must rapidly switch from an eccentric contraction to a concentric contraction (stretch-shortening cycle) [31–33] while under the influence of the contractile activity of the quadriceps femoris muscle. Therefore, neuromuscular coordination appears to play an important role in this activity. A hamstring muscle injury is conjectured to occur in the event of a muscle dyssynergy, such as a disorder in the timing of the contraction from the late swing phase to the early contact phase [14].

In sprinting, many muscular activities relating to multiple joints undergo synergic control during an extremely short period and their execution depends on the speed level. The central nervous system plays a key role in executing the motion characterized by a smooth contraction. During supramaximal running, it is possible to achieve a higher stride frequency than with maximal running. However, the excessive stride frequency increases the possibility of hamstring injury [19]. Mero and Komi [1,10,18] have already shown that adequate neuromuscular performance is important for supramaximal running. Moreover, it has been suggested that certain neural networks for controlling locomotion are needed specifically for locomotor speeds [34]. Thus, appropriate neural control is a key factor in preventing hamstring injury during supramaximal running for a higher stride frequency.

Agility training is practiced as a means of learning the rapid motion needed to cope with a supramaximal level of running. Sprinters who practiced using ladders and mini hurdles exhibited rapid stepping equivalent to or faster than the stride frequency that had been observed during supramaximal running [19]. Motion training at a high level, such as supramaximal running, which requires a quick sprinting motion, incorporates the learning of new muscle recruitment patterns that involve peripheral sensory input. It is likely that the muscle synergy adapted to high-level sprinting motion was acquired by all sprinters through the use of ladders and mini hurdles. Thus, the number of hamstring injuries decreased during supramaximal running.

### 4.4. Effects of Strength Training on Hamstring Injury Incidence

The contribution of the hamstring to sprint speed increases with running velocity [2]. In sprinting, the function of the hamstring in the late swing phase is that of the hip extensor concentrically acting to quickly swing the thigh back [2,4] while also acting as a knee flexor to eccentrically decelerate the forward swing of the lower leg [2,4,31]. In the early contact phase, the hamstring function minimizes the loss of running speed by performing a concentric action as the knee flexor and hip extensor muscles, enabling the body's center of gravity to shift forward smoothly [4,31]. The hamstrings must generate a large amount of power during these phases to maximize running speed during sprinting [2–4]. The relative iEMG of the biceps femoris is significantly greater in the late swing phase during supramaximal running than it is in running at maximum speed [5]. The power demonstrated by the eccentric contraction of the knee flexors (hamstrings) and that demonstrated by the concentric contraction of the hip extensor in the late part of the swing phase are reported to increase during supramaximal running [5]. Supramaximal running, wherein a sprinter sprints at a supramaximal level by artificial means, forces the hamstring to perform activity at extremely intense levels [5,6]. In sprinters, these generated high forces are postulated to be related to the hamstring injuries during supramaximal running.

A relationship between an insufficient hamstring strength in eccentric contraction and hamstring injury occurrence has been reported [15,25,32,35]. Sugiura et al. [15] suggested that hamstring injury in elite sprinters was associated with weakness of the hamstrings in eccentric action and weakness of the hip extensors in concentric action. Therefore, strengthening training was undertaken to increase muscle strength so much so that it can respond to the high-intensity (force) activity demonstrated by the hamstring in supramaximal running through the exercise of contracting concentrically and eccentrically.

In the 2000s, several studies have demonstrated the effectiveness of eccentric strength training on hamstring muscles [20,21,36,37]. Systematic reviews [38] have recommended that Nordic hamstring exercises is effective to prevent hamstring muscle injury. Nordic hamstring exercises can put the maximum load on the hamstring during the eccentric phase. The primary injury mechanism is high-speed running; more specifically, it occurs during the swing phase, when the hamstrings are required to contract forcefully while lengthening to decelerate the extending knee and flexing hip [39]. Therefore, the low level of hamstring eccentric strength [15,25,32,35] has been cited as the primary risk factor for hamstring injury. The Nordic hamstring exercise can maximize loading on the hamstring in the eccentric phase [21]. During the execution of the Nordic hamstring exercise, a progressive resistant torque imposed by the trunk position leads to increased neuromuscular hamstring activation at longer muscle lengths [40,41], precisely at the position most vulnerable to injury [42]. The lumbopelvic region muscles effect hamstring injury risk more than the distal muscles of the knee and ankle do [15].

The practice of strength training applies an eccentric load to the hamstring through the trunk position at the injury position. Eccentric strength training worked well against load on hamstring during supramaximal training. In this study, the number of hamstring injuries during the intervention period with eccentric strength training decreased relative to the baseline period. All sprinters strengthened their hamstrings with leg curls, hip extensions, and two types of modified Nordic hamstring exercises. This likely decreased the number of hamstring injury cases during supramaximal running.

### 4.5. Effects of Flexibility Training on Hamstring Injury Incidence

When hamstring flexibility decreases, the length of the hamstring during the demonstration of peak torque is reduced [43]. Further, Brockett et al. [36] proved that short muscle length during the demonstration of peak torque is a risk factor for the occurrence of a hamstring injury. Flexibility is an intrinsic property of body tissue and determines the range of motion achievable without injury at a joint or group of joints [44]. Increase in the joint range of motion is associated with decrease in passive resistance to stretching. This decrease in resistance is called decrease in muscle stiffness or increase in muscle

compliance [45]. Increased passive stiffness of the hamstrings and decreased knee ROM are both risk factors for hamstring injury [46,47].

The purpose of stretching before an athletic event or training is to ensure that the athlete has a sufficient range of motion in his joint to optimally perform the athletic activity and to decrease muscle stiffness or increase muscle compliance. Therefore, stretching is used to affect both injury risk and performance [24,45].

However, the practice of static stretching has been demonstrated to have a potentially negative impact on performance that directly follows the stretching [48,49]. In contrast, dynamic stretching has been shown to have a positive effect on performance; it has recently been endorsed as a replacement for static stretching [50,51]. It is recommended to do dynamic stretching as rapidly as possible before the performance [52]. Dynamic stretching should be part of warmup routines because of its similarity to the movement patterns of the subsequent activity, in addition to the promotion of greater muscle activation [52].

Performing dynamic stretching, the neuromuscular system is allowed a softer musculotendinous system with increased length and the ability to perform larger movements [53]. The objective of training with dynamic stretching is to acquire flexibility in the lumbopelvic region muscles and to adapt the hip joint to a mobile state where dynamic flexibility is secured. The hip joint movement adapted to supramaximal running is likely acquired by stretching the hamstring, quadriceps femoris, and other muscles while actively moving the joints.

Dynamic stretching is conjectured to function effectively in preventing hamstring injury while also assisting athletes to perform at a high level. The practice of dynamic stretching is potentially effective as a method of preventing hamstring injuries in supramaximal running. However, Opplert and Babault [54] found that few studies had been conducted regarding the methodological rules and effects of dynamic stretching; hence, future studies are hoped for in this area.

### 4.6. Limitation and Implication

Having conducted three hamstring injury training programs at the same time—namely, in agility, strength, and flexibility—we are unable to demonstrate their relative effects on injury reduction. Therefore, it is necessary to investigate what training or combination is the most effective.

Preventing hamstring injury is just as important as improving performance for athletes, although spending time on prevention training can be impractical and difficult. With more performance training comes better prevention of hamstring injury.

### 5. Conclusions

The disadvantage of tow-training is the occurrence of hamstring injury, which can be corrected for by combination of programs aimed at strengthening eccentric muscles, improving the neuromuscular system, and increasing dynamic flexibility. To ensure effectiveness of tow-training—that is, a high stride frequency—sprinters are encouraged to press the leg onto the ground as fast as possible, resulting in preventing the stride lengthening as well.

**Author Contributions:** Conceptualization, Y.S. and K.S. (Kazuhiko Sakuma); Methodology, Y.S. and K.S. (Kazuhiko Sakuma); Software, S.F.; Validation, K.S. (Kazuhiko Sakuma) and Y.S.; Formal analysis, S.F.; Investigation, K.S. (Kazuhiko Sakuma) and Y.S.; Resources, K.S. (Kazuhiko Sakuma) and K.S. (Keishoku Sakuraba); Data curation, K.S. (Kazuhiko Sakuma); Writing—original draft, Y.S. and K.S. (Kazuhiko Sakuma); Preparation, Y.S. and K.S. (Kazuhiko Sakuma); Writing—review and editing, Y.S. and S.F.; Visualization, Y.S., K.S. (Kazuhiko Sakuma), and S.F.; Supervision, K.S. (Kazuhiko Sakuma) and K.S. (Keishoku Sakuraba); Project administration, K.S. (Keishoku Sakuraba). All authors have read and agreed to the published version of the manuscript.

**Funding:** This research received no external funding.

**Institutional Review Board Statement:** The human ethics committee of Juntendo University approved the protocol for this research (protocol approval code 27–13, approved on 17 November 2005).

**Informed Consent Statement:** Informed consent was obtained from all subjects involved in this study.

**Data Availability Statement:** Raw data are not open to the public.

**Conflicts of Interest:** The authors declare no conflict of interest.

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
