# Peer review of "Hamstring Injury Prevention Program and Recommendation for Stride Frequency during Tow-Training Optimization"

_applsci, doi:10.3390/app11146500_

Round 1
Reviewer 1 Report
This article is not clearly aligned with the purpose, methods, conclusion, or title. Make sure it is clear what you are researching. I do not know if you are looking solely at tow training, or tow training in conjunction with strengthening/stretching, or stride frequency, or hamstring injuries.
Make sure to identify and describe tow training earlier, I am an endurance athlete and have worked with a lot of runners/track athletes as a physical therapist and I was still unclear that this was "assisted" running instead of weighted running/towing. Perhaps I am still unclear.
Many statements in here are lacking citations. There are large jumps from the research to applying it to your study. Make sure the research supports all of your statements and there are no large assumptions.
This article is lengthy with long descriptions of strengthening and stretching. Be concise and condense this background information. Much of it is unnecessary and again, there are large jumps/assumptions between the research and application to your study regarding stretching and strengthening

Author Response
Thank you for your review comment.
=for the first comment=
The main focus of this article is hamstring injury prevention. The point was not clearly mentioned throughout the article consistently in the purpose, methods, conclusion, and title. Thus, I have made the point of the article clear; the title was changed and the other sections were minor changed to align focusing on increasing stride frequency while decreasing hamstring injury.
=for the second comment=
As you’ve known, supramaximal running is assisted. To make clearer, I put extra explanation at the beginning in the Introduction.
In figure 2, the one on left-hand is omitted to avoid confusion.
=for the third and fourth comments=
* I’ve added citations.
* I’ve edited article.
Based on your comments, I’ve worked with co-authors.
Your detailed comments gave us, authors, a chance to study deeper. I, as a correspondence author, appreciate for your support.
Yusaku Sugiura
Reviewer 2 Report
In the limitation section you indicate that it is not necessary to waste time only on prevention programs. This statement seems risky to me, since although more training improves injury prevention, a more in-depth study would have to be made on the different prevention programs and systems. Perhaps programs directed by double graduates in Sciences of Physical Activity and Sports and Physiotherapy in prevention can be just as useful as training.
Perhaps I would try not to sentence this statement only with the result obtained, since it seems very risky to me.
Author Response
Thank you for your review comment.
It seem that my sentence was exaggerated what I wanted to address. The point I wanted to focus was sprinters and coached did not need to spend time and energy solely "prevention" but "practice." However, since my sentence has an inappropriate word, I've decided to delete the sentence
Yusaku Sugiura, Correspondence Author
Reviewer 3 Report
Line 28: add a reference
Lines 32-22: add a reference for the first part of the sentence
Lines 34-36: add a reference
Lines 44-45: add a reference
Line 49:….,as Hicks reports,
Lines 57-58: add a reference
Lines 67-69: add a reference
Line 79: what is the hypothesis of your study?
Line 82: how were the subjects recruited and what were the criteria for inclusion and exclusion
Line 86: Olympic Games
Line 185: have you thought of radiological examinations?
Line 217: the discussion reads like a review; I suggest starting with the main findings and to compare with your hypothesis/hypotheses to check whether you can confirm or not. This would give a structure to the discussion
Author Response
Thank you for your review comment.
In response to your feedback, "Line 217" issue, I have made changes.
Fist of all, I've add a section at the beginnig of the discussion section. In this section, briefly findings are explained. In each section, each finding is deeply examined so as not to merely review the issue.
For the rest of the issue you pointed out, I've followed your suggestions.